# A meta-analysis of circulating microRNAs in the diagnosis of papillary thyroid carcinoma

Yan liu[1]⊛, Houfa Geng[2]⊛, Xuekui Liu[2], Mingfeng Cao[3]*, Xinhuan Zhang[3]*

**1** Shandong First Medical University & Shandong Academy of Medical Science, Tai'an, China, **2** Department of Endocrinology, Xuzhou Central Hospital, Affiliated Hospital of Medical School of Southeast University, Xuzhou, China, **3** Department of Endocrinology, The Second Affiliated Hospital of Shandong First Medical University & Shandong Academy of Medical Science, Tai'an, China

⊛ These authors contributed equally to this work.
* kathy0418@163.com (XZ); 18653192961@163.com (MC)

**Data Availability Statement:** All relevant data are within the paper and its Supporting information files.

**Funding:** This work was supported by the Higher Educational Science and Technology Program of

## Abstract

### Background

Aim of this meta-analysis was to evaluate the overall diagnostic value of circulating mini miRNAs for papillary thyroid carcinoma (PTC) and to find the possible molecular marker with higher diagnostic value for PTC.

### Methods

We searched the Pubmed, Cochrane and Embase database until June 2020. We selected relevant literatures associated with the diagnosis of PTC with circulating miRNAs. The number of cases in experimental group and the control group, sensitivity and specificity could be extracted from the literatures.

### Results

We got 9 literatures including 2114 cases of PTC. Comprehensive sensitivity was 0.79, comprehensive specificity was 0.82, positive likelihood ratio was 4.3, negative likelihood ratio was 0.26, diagnostic advantage ratio was 16. The summary receiver operating characteristic curve was drawn and the Area Under the Curve was 0.87.

### Conclusions

Circulating microRNAs may be promising molecular markers for the diagnosis of papillary thyroid carcinoma. Combined detection of certain serum microRNAs can improve the diagnostic accuracy of papillary thyroid carcinoma. Especially MiR-222 and miR-146b may be prime candidates for the diagnosis of PTC in Asian population.

Shandong Province, China (Grant No. J17KA246); Natural Science Foundation of Shandong Province of China (Grant No. ZR2017LH023); Medical and Health Project of Shandong Province, China (Grant No. 2015WS0096), Science and Education Project for Young Medical Talents, Jiangsu Province, China (Grant No.QNRC2016388), as well as Academic Promotion Program of Shandong First Medical University (Grant No. 2019QL017).

**Competing interests:** The authors have declared that no competing interests exist.

## Introduction

Thyroid cancer is a kind of common malignant tumor. The incidence of thyroid cancer ranks sixth among all malignant tumors [1]. Papillary thyroid carcinoma (PTC) is the most common histological type of thyroid cancer. To date the most reliable method to evaluate thyroid nodules is fine needle aspiration cytology (FNA), but the FNA still has its technical limitations. Less than 20% of surgically removed nodules are malignant [2–4]. Therefore, in order to improve the diagnostic accuracy of PTC and avoid unnecessary surgery, more and more researchers focused their attention to explore the molecular markers of PTC. Recent studies have shown that the abnormal expression of microRNAs (miRNAs) is closely related to the occurrence and development of tumors [5–7]. Studies have reported that thyroid cancer is associated with somatic cell mutation [8], gene expression [9], and miRNA expression [10, 11]. Many miRNAs may play an important role in the occurrence, development, metastasis and prognosis of thyroid cancer [10, 11].

MicroRNAs are a class of endogenous, small (21–23 nucleotides in length), non-coding RNAs. They can negatively regulate gene expression and regulate basic physiological processes, such as cell differentiation, growth, proliferation, metabolism and apoptosis [12, 13]. Circulating miRNAs are stable in circulation and can be detected stably from peripheral blood. These properties of miRNAs make them promising non-invasive biomarkers for cancer and other diseases. However, to date there are few studies on circulating miRNAs in PTC, and the conclusions are inconsistent. The diagnostic value of circulating miRNAs has not been accurately evaluated.

In this study, we hope to evaluate the overall diagnostic value of serum/plasma miRNAs for PTC by analyzing the research results in the existing literature and using statistical software, and try to find sensitive and specific molecular markers for the diagnosis of PTC.

## Materials and methods

### Literature search strategy

By searching the pubmed, Cochrane and Embase database, we searched for the literatures related to the diagnosis of thyroid papillary cancer by serum miRNA, and restricted the publication date of literature by the end of 30. Jun. 2020. Search terms were "thyroid cancer", "papillary thyroid carcinoma", "thyroid nodule", "microRNA", "miRNA", "miR", "serum", "circulation", "plasma". The English retrieval method is {("thyroid cancer" OR "papillary thyroid carcinoma" OR "thyroid nodules") AND ("microRNA" OR "miRNA" OR "miR") AND ("serum" OR "circulation" OR "plasma")}.

### Selection criteria

Inclusion criteria include: (1) Studies associated with the diagnosis of PTC by serum/plasma miRNAs. (2) The experimental groups were PTC, and the control groups were benign thyroid disease (including nodular goiter or thyroid adenoma). The age and sex of these groups were balanced and comparable. (3) The number of cases, sensitivity and specificity could be extracted from the experimental group and the control group, and table data could be calculated, including true positives (tp), false positives (fp), false negatives (fn), true negatives (tn). (4) The results of pathological examination after operation were taken as the gold standard for diagnosis. (5) The detection methods and reagent sources of miRNAs in the literature were clear. (6) No limitation on age, sex, nationality and race. The language of the literatures was English. Exclusion criteria were as follows: (1) Literature review, meetings and letters; (2)

Simple descriptive study without control group; (3) Repeated literature; (4) Literatures that could not be obtained complete data.

## Data extraction and management

According to inclusion and exclusion criteria, two evaluators screened literatures strictly and extracted data independently. The main data extracted from the study were: the first author, the research country, the publication time, the target miRNAs, the sample source of miRNAs, the four-grid table data (including tp, fp, fn, tn). If the results of the two evaluators were inconsistent, they would cross-check the data.

## Quality assessment

QUADAS-2(Quality Assessment of Diagnostic Accuracy Studies-2) in Revman 5.3 was used to evaluate the quality of the included studies, include Patient Selection, Index Test, Reference Standard, Flow and Timing. Significant issues included "yes", "no", "uncertainty". Risk level of bias are divided into "high", "low", "uncertainty". Clinical applicability of selected patients, index testing and standard of reference were evaluated respectively. Two evaluators independently conducted quality evaluation, and finally cross-checked the relevant data extracted. If there were differences, they would consult the mentor.

## Data synthesis and statistical analysis

The statistical software used was STATA 15.1 and Meta-disc 1.40. Firstly, the heterogeneity of the study was tested by $I^2$ test: if $I^2 \leq 50\%$ used fixed effect model, if $I^2 \geq 50\%$ used random effect model. Statistical analysis was performed including sensitivity, specificity, positive likelihood ratio, negative likelihood ratio and diagnostic odds ratio. The comprehensive working characteristic curve was drawn and the area under curve (AUC) was calculated. Additionally, the value of AUC is 0.5–0.7, 0.7–0.9, and 0.9–1.0, which represents the low, moderate, and high diagnostic efficacy, respectively. The Spearman correlation coefficient was calculated by metadisc1.40 software to evaluate the heterogeneity caused by threshold effect: if $P > 0.05$, there is no threshold effect, and all indicators can be combined; if $P < 0.05$, there is threshold effect, and the research indicators can not be combined, but simply described. Then, subgroup analysis and regression analysis were used to explore the source of heterogeneity. Sensitivity analysis was conducted to explore the robustness and reliability of the selected literatures. Finally, the Deek funnel plot was used to test the publication bias. If $P < 0.10$, there was publication bias.

# Results

## Summary of searches and study selection process

Through searching the database, 215 literatures were obtained, of which 77 literatures remained after excluding duplicated literatures. By simply reading the title and abstract, we excluded 17 literatures of reviews, conference and literatures by the same author. Following further evaluation, 51 literatures were excluded for the following reasons: literatures focusing on the prognosis and therapy (n = 13), the mechanism of occurrence (n = 12), non-serum/plasma samples (n = 18), and literatures that could not be extracted complete data (n = 8). Finally, we got 9 literatures [14–22], including 29 unique studies (part of the literatures included two or more independent studies). A total of 2114 cases of papillary thyroid carcinoma and 1615 cases of control group were included (Fig 1).

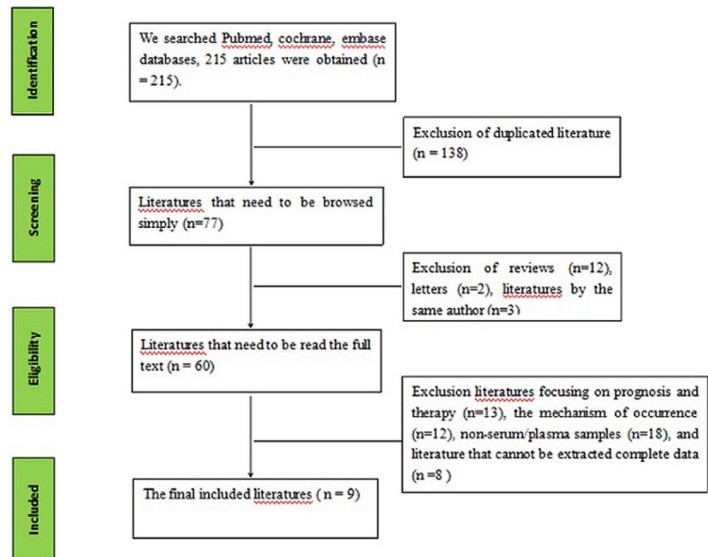

**Fig 1. Flow diagram of screening studies.**

## The basic characteristics of literature

The basic features of the literatures were listed in S1 Table. We extracted the main data from the study, including the first author, the research country, the publication time, the target miR-NAs, the sample source of miRNAs, the true positives, false positives, false negatives, true negatives.

## Quality evaluation results

Revman 5.3 was used to evaluate the quality of 9 articles (Fig 2). In general, the quality of the included literature was barely satisfactory. Most of the studies were case-control studies, and the reference diagnostic gold standard of all experiments was the pathological results after thyroid cancer surgery. However, Rosignolo [19] did not mention appropriate selection and exclusion criteria, which led to the risk of bias in the field of case selection is not clear. We can also see from the quality assessment diagram that the bias risk of the test to be evaluated is high, which may be related to the implementation of the test, the different process and the lack of preset threshold. Because there are few studies on the diagnosis of PTC by circulating miR-NAs, most of them are based on the ROC curve to obtain the optimal diagnostic threshold, so there is no unified diagnostic threshold standard. Finally, Yu et al [14], Cantaras et al. [15], Li et al. [17] and Yu et al. [18] did not observe the expression of circulating miRNA in postoperative PTC patients, which may cause uncertainty in the part of flow and timing.

## Statistical analysis results

The meta-analysis was carried out by using the Midas module in Stata 15.1. The $I^2$ of sensitivity and specificity were 90.25 and 83.04. On account of the $I^2$ more than 50%, we selected the random effect model in Stata to obtain meta-analysis results of circulating miRNAs in the diagnosis of PTC: comprehensive sensitivity was 0.79 [95% CI: 0.72, 0.84], comprehensive specificity was 0.82 [95% CI: 0.76, 0.86], positive likelihood ratio was 4.3 [95% CI: 3.2, 5.7], negative likelihood ratio was 0.26 [95% CI: 0.20, 0.35], diagnostic advantage ratio was 16 [95% CI: 10, 25]

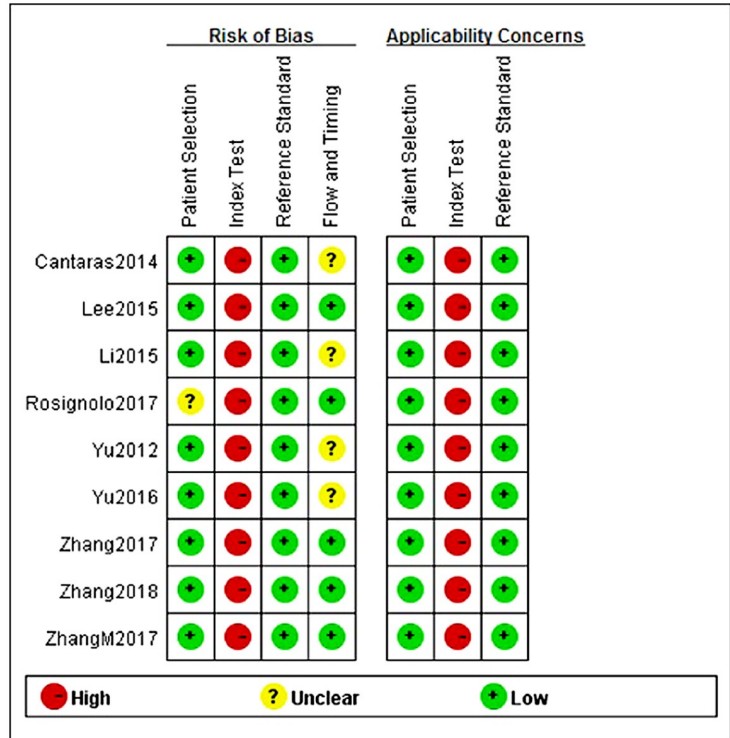

**Fig 2. Quality assessment diagram.**

(Fig 3). The summary receiver operating characteristic curve (SROC) was drawn (Fig 4) and the Area Under the Curve (AUC) was 0.87 [95% CI: 0.84, 0.90].

It was concluded that circulating miRNAs could be used as valuable molecular markers for the diagnosis of papillary thyroid carcinoma. The Spearman correlation coefficients of sensitivity logarithm and (1-specificity) logarithm (r = 0.146, p = 0.451) were calculated by metadisc 1.40, which showed that there was no obvious threshold effect between the studies.

## Subgroup analysis

Considering the heterogeneity was considerable in our meta-analysis with $I^2$ more than 50%, subgroup analyses and meta-regression analysis were performed. The subgroup analyses was carried out according to the study area, the number of miRNAs, the types of miRNAs, the source of miRNAs samples and internal reference. We found the following results:(1) Circulating miR-222 and miR-146b may be valuable molecular markers of papillary thyroid carcinoma in Asian population. Among them, the diagnostic value of miR-222 was sensitivity: 0.60, 95% CI: 0.57–0.79; specificity: 0.90, 95% CI: 0.84–0.94; PLR: 6.8, 95% CI: 4.1–11.4; NLR: 0.34, 95% CI: 0.24–0.49; DOR: 20, 95% CI: 10–42; AUC: 0.90, 95% CI: 0.84–0.94, with high diagnostic efficacy (Fig 5a). The diagnostic value of miR-146b was sensitivity: 0.82, 95% CI: 0.60–0.93; specificity: 0.71, 95% CI: 0.60–0.79; PLR: 2.8, 95% CI: 1.8–4.2; NLR: 0.26, 95% CI: 0.10–0.66; DOR: 11, 95% CI: 3–38; AUC: 0.72, 95% CI: 0.68–0.76, with medium diagnostic effect (Fig 5b). However, there are few studies on the diagnosis of PTC by these two miRNAs in circulation. We only included three such literatures. There are 270 PTC patients in miR-222 and 234 patients in miR-146b. The included studies are few and the number of cases is not rich. Therefore, it can only provide a clue for the future research direction. At the same time, we need to

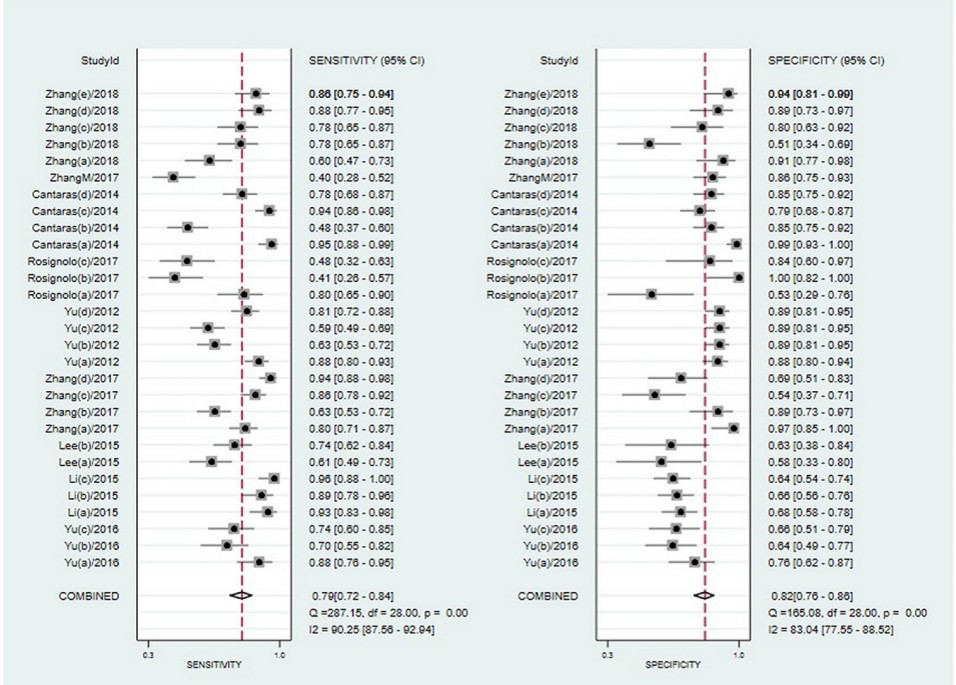

**Fig 3. Forest plot of sensitivity and specificity of circulating microRNAs in the diagnosis of papillary thyroid carcinoma.**

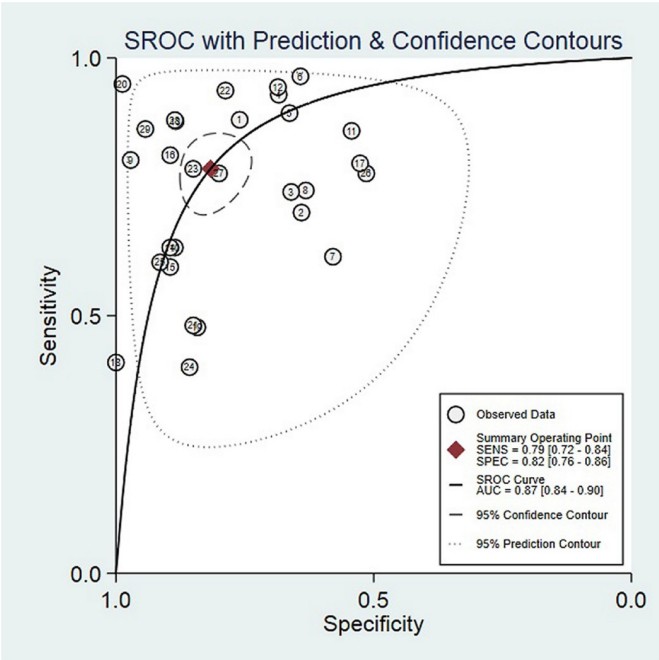

**Fig 4. The summary receiver operating characteristic curve of circulating microRNAs in the diagnosis of papillary thyroid carcinoma.**

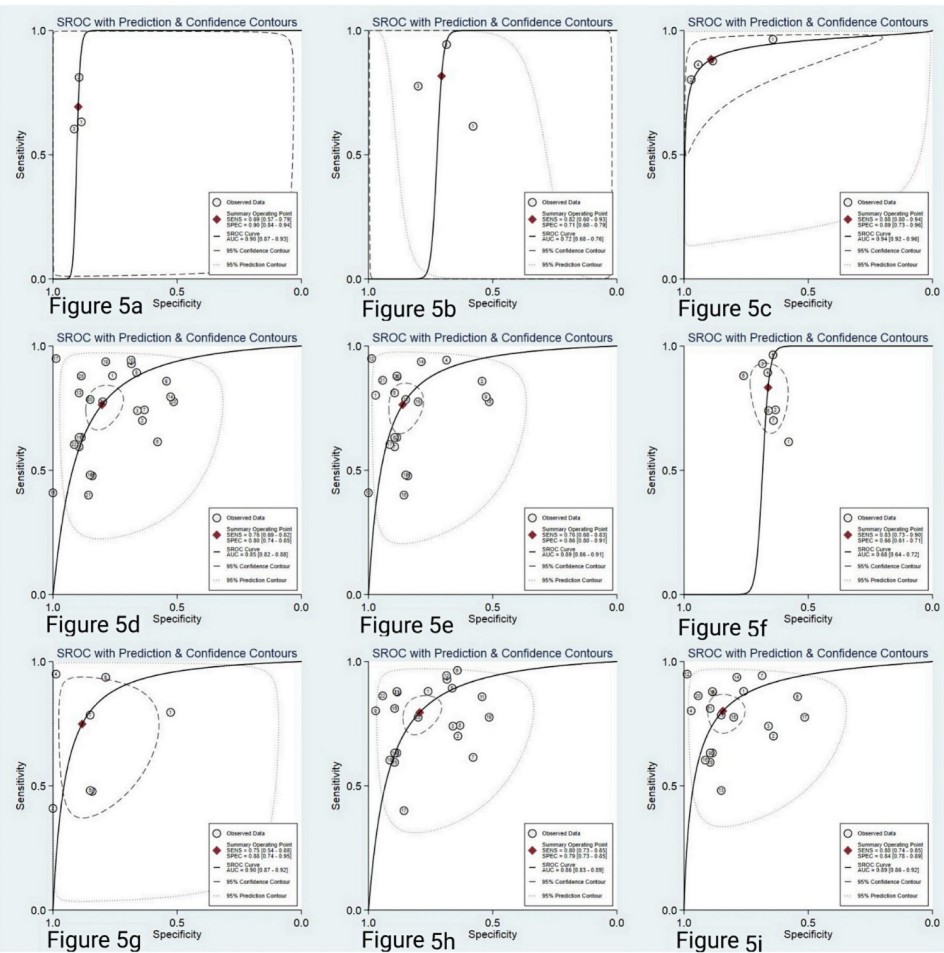

**Fig 5. SROC curve showed the diagnostic efficacy of miR-222 (a) and mir-146b (b), miRNA combination (c) and single miRNA (d), serum miRNA (e) and plasma miRNA (f), Asia (g) and Europe (h), and mir-16 as internal reference (i).**

further expand the sample size to verify in the future research. However, in European population, there is no relevant literature on circulating miR-222 and miR-146b for papillary thyroid carcinoma, so the diagnostic value of these two miRNAs in European population is still unclear. (2) The combination miRNA (sensitivity: 0.88, 95% CI: 0.80–0.94; specificity: 0.89, 95% CI: 0.73–0.96; PLR: 8.1, 95% CI: 3.2–20.7; NLR: 0.13, 95% CI: 0.08–0.21, DOR: 62, 95% CI: 29–136; AUC: 0.94, 95% CI: 0.92–0.96 (Fig 5c), were more valuable than single miRNA (Fig 5d). (3) It is noteworthy that serum miRNAs (AUC: 0.89, 95% CI: 0.86–0.91, Fig 5e) may be more conducive than plasma miRNAs (AUC: 0.68, 95% CI: 0.64–0.72, Fig 5f) in the diagnosis of papillary thyroid carcinoma. (4) Race also has an impact on the diagnostic value of miRNAs for papillary thyroid cancer. The diagnostic value of Europeans (AUC: 0.90, 95% CI: 0.87–0.92, Fig 5g) may be higher than that of Asians (AUC: 0.86, 95%CI: 0.83–0.89, Fig 5h). (5) The sensitivity, specificity, PLR, NLR, DOR and AUC of miR-16 were 0.80, 0.84, 5.1, 0.24, 22 and 0.89 (Fig 5i), which suggests that miR-16 as an internal reference is more helpful than other miRNAs. The comprehensive diagnostic indexes of each subgroup are shown in S2 Table. For regression analysis (Fig 6), we found that the region (P<0.001) and sample source (P<0.01) were the main sources of heterogeneity.

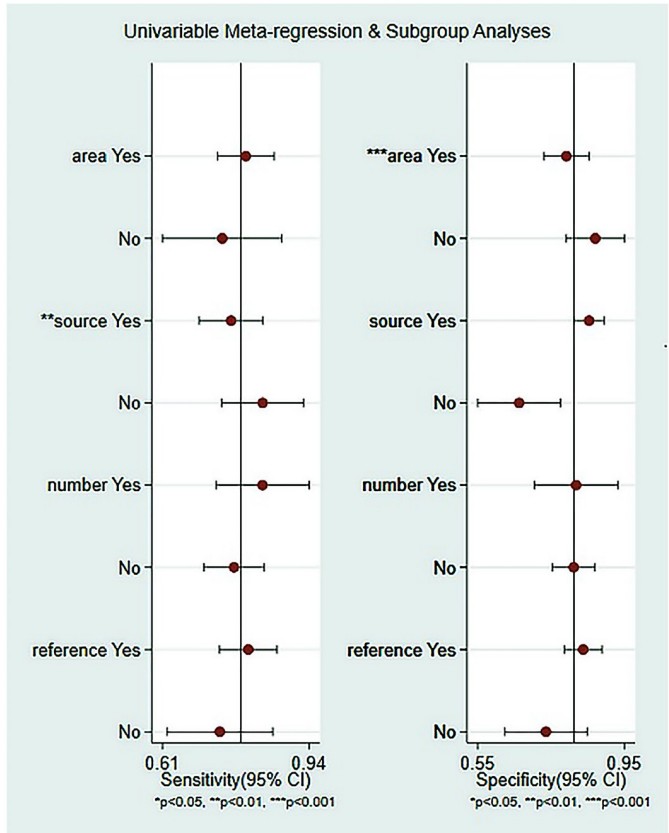

**Fig 6. Regression analysis.**

Sensitivity analysis of the included studies was performed (Fig 7). After each exclusion of a single study, the remaining studies were recombined, and the results did not change significantly, which indicated that the sensitivity of the included studies was low, and the results were more robust and credible.

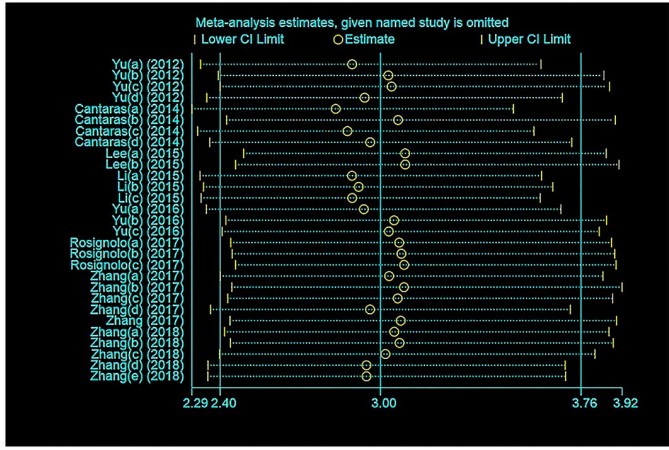

**Fig 7. Sensitivity analysis.**

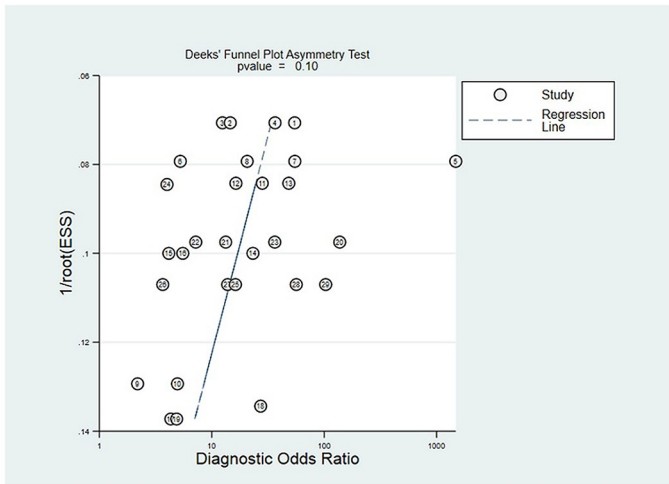

**Fig 8. Deek funnel.**

## Publication bias test

Stata 15.1 was used to plot Deek funnel plot. As displayed in Fig 8, it had some asymmetry, especially in the 20th study. The p value we calculated is 0.10, which indicated a certain publication bias in the included studies.

## Discussion

Papillary thyroid carcinoma (PTC) is the most common histological type of thyroid cancer. The incidence of PTC is increasing year by year, and the number of asymptomatic thyroid micro-carcinomas (tumors with a maximum diameter less than or equal to 1 cm) is increasing.

To date the most reliable method to evaluate thyroid nodules is fine needle aspiration cytology (FNA), but the FNA still has its technical limitations. There are still exist false positive and false negative in FNA. About diagnose 20%–30% of thyroid nodules can not determined through FNA, especially for Bethesda III and IV follicular nodules, FNA can not identify the nature of nodules, which may led to the surgical treatment of some nodules that did not need surgical resection.

MicroRNAs (miRNAs) are highly conserved non-coding single-stranded small RNA molecules, which participate in cell proliferation, differentiation and apoptosis [23]. Post-transcriptional regulation mediated by microRNAs is an important way to regulate gene expression. The mutation or abnormality of microRNAs plays a key role in the occurrence, development, invasion and metastasis of tumors [24], and it is closely related to the therapeutic response, staging, prognosis and recurrence detection of tumors [25]. Because of the good stability of miRNAs in peripheral blood, the detection of miRNAs is noninvasive, sensitive and simple, which has become a hot topic in molecular biology of tumors in recent years [26]. Increasing studies have found that circulating microRNAs can be used as molecular markers for a variety of diseases especially in malignant tumor. Our study also found that miR-222 and mir-146b have high specificity in the diagnosis of papillary thyroidcarcinoma. So Combined detection of microRNA in circulating and cytopathology specimens may greatly improve the diagnostic accuracy of thyroid cancer.

Our study mainly discussed the diagnostic value of circulating miRNAs in PTC. Meta analysis showed that the total sensitivity was 79%, and the total specificity was 82%. The positive likelihood ratio was 4.3, which showed that the positive rate of circulating miRNAs in PTC patients was 4.3 times higher than that in patients with benign group. The overall diagnostic ratio was 16 as well as the negative likelihood ratio was 0.26, suggesting that circulating miR-NAs may have a good diagnostic ability for PTC. Commonly, the area under the ROC curve was used to estimate the accuracy of diagnosis, ROC curve area at 0.7 to 0.9 indicating the index has a higher accuracy; area above 0.9 indicating the index has the highest accuracy. In our study, the AUC was 0.87, which showed that circulating miRNAs may have a promising diagnostic value for PTC.

In our study, 20 kinds of microRNAs were included. Several kinds of microRNAs have been studied many times, including miRNA-146b, miRNA-221, miRNA-222. Subgroup analysis of microRNA type showed that miRNA-222 and miRNA-146b had high diagnostic value, especially the miRNA-222. The sensitivity, specificity and diagnostic ratio of miRNA-222 were 0.70, 0.90 and 22.55 respectively. MiRNA-222, locates on chromosome Xp11.3 [27], it is a well-defined proto-oncogene family. It is abnormally expressed in many tumors, such as primary hepatocellular carcinoma and pancreatic cancer [28]. In this study, we only included three literatures about the relationship between miRNA222 and papillary thyroid carcinoma. The number of literatures was limited, which need to be verified by high-quality and larger sample studies. From the subgroup analysis, we could also find that the diagnostic value of microRNAs in serum samples may be higher than that in plasma samples for PTC, which indicated that we could pay more attention to serum samples in the future. Compared with plasma, the absence of some coagulation factors and fibrinogen in serum may reduce the interference factors in the determination of circulating microRNAs. In addition, some studies have found that a large number of cytokines can release during the process of activation and destruction of platelets, which may affect the determination results. However, there are few literatures discuss it, and more further researches are needed to explore the expression differences between serum and plasma microRNAs and their possible reasons and significance [29].

Subgroup analysis of the number of miRNAs showed that sensitivity, specificity and diagnostic ratio of the combined miRNAs were 0.87, 0.82, 64.58, which was higher than that of single miRNAs in the diagnosis of PTC. So it is more important to pay more attention to the specific combined detection of miRNAs than single circulating miRNA, in order to better diagnosis of PTC and reduce unnecessary surgery.

At present, real-time fluorescent quantitative (RT-qPCR) has been widely used in the studies of the mRNA, the detection of DNA, the determination of single nucleotide polymorphism and so on [30, 31]. Due to the advantages of it such as high sensitivity, good repeatability, accurate quantification and so on, it is also widely used in the detection of circulating miRNAs. Among the included literatures, some literatures initially used gene microRNA array to screen miRNAs differently expressed between groups, followed by qPCR amplification after reverse transcription into cDNA. The development of RT-qPCR has accelerated the development of miRNAs in cancer diagnosis and treatment.

Limitations of our study include the following aspects: Firstly, the bias risk of the index test in the literature was higher, which may be related to the different method of the study design, the implementation process of test and the absence of pre-set threshold. To date, there are few studies on the diagnosis of PTC by miRNAs. Most of the studies got the best diagnostic threshold based on the ROC curve. There was no unified diagnostic threshold standard, so the bias risk of the index test was higher. Secondly, the meta-analysis had great heterogeneity. We had not found the source of heterogeneity through subgroup analysis. The possible reasons for the

heterogeneity may be as follows: age, sex, number of lesions, capsular invasion, lymph node metastasis and stage of papillary thyroid carcinoma in the experimental group were different; design, detection methods and reagent selection of different studies were inconsistent; the cut-off values used were different. Thirdly, we could see from the funnel plot that it had certain asymmetry, especially in the 20th study. We review the source of the literature: Silvia Cantara et al. have identified for the first time of two miRNAs differently expressed in serum of PTC patients in a caucasian population. They found that the best diagnostic accuracy was miRNA95 with a sensitivity of 94.9% and a specificity of 98.7%. Its sensitivity and specificity were significantly higher than any other microRNAs we included, which may be the reason why it deviated from the reference line on the funnel plot. Therefore, this requires us to further expand the sample size in future research to verify the diagnostic value of microRNA95 for PTC in a Caucasus population. Overall, we thought the result of publication bias testing was still acceptable.

## Conclusion

Circulating microRNAs may be promising molecular markers for the diagnosis of papillary thyroid carcinoma. Especially MiR-222 and miR-146b may be prime candidates for the diagnosis of papillary thyroid carcinoma in Asian population,. Combined detection of certain serum microRNAs can improve the diagnostic accuracy of papillary thyroid carcinoma.

## Supporting information

**S1 Table. The basic characteristics of inclusion literatures.**
(PDF)

**S2 Table. Subgroup analysis of circulating microRNAs in the diagnosis of PTC.**
(PDF)

**S1 File. Prisma checklist.**
(PDF)

**S2 File. Search strategy and search terms of this manuscript.**
(PDF)

## Author Contributions

**Formal analysis:** Mingfeng Cao.

**Funding acquisition:** Xinhuan Zhang.

**Investigation:** Xinhuan Zhang.

**Methodology:** Xuekui Liu.

**Project administration:** Xinhuan Zhang.

**Supervision:** Mingfeng Cao, Xinhuan Zhang.

**Writing – original draft:** Yan liu.

**Writing – review & editing:** Houfa Geng.

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
