## [Decision Letter · Decision Letter 0]

1 Feb 2021

PONE-D-20-40536

A meta-analysis of circulating microRNAs in the diagnosis of papillary thyroid carcinoma

PLOS ONE

Dear Dr. Huan Xin Zhang,,

Thank you for submitting your manuscript to PLOS ONE. After careful consideration, we feel that it has merit but does not fully meet PLOS ONE’s publication criteria as it currently stands. Therefore, we invite you to submit a revised version of the manuscript that addresses the points raised during the review process.

We look forward to receiving your revised manuscript.

Kind regards,

Giancarlo Troncone

Academic Editor

PLOS ONE

Journal Requirements:

2. At this time, we ask that you please provide the full search strategy and search terms for at least one database used as Supplementary Information.

3. Thank you for including the statement that "The database search was conducted on XXXXXX". Please revise this statement to clarify whether all databases were searched from inception, or if there were any limits placed on the publication dates in your search.

4. Please ensure you have thoroughly discussed any potential limitations of this study within the Discussion section, including the potential impact of confounding factors.

5.We note that you have indicated that data from this study are available upon request. PLOS only allows data to be available upon request if there are legal or ethical restrictions on sharing data publicly. For information on unacceptable data access restrictions, please see http://journals.plos.org/plosone/s/data-availability#loc-unacceptable-data-access-restrictions.

6.We note that the grant information you provided in the ‘Funding Information’ and ‘Financial Disclosure’ sections do not match.

7.PLOS requires an ORCID iD for the corresponding author in Editorial Manager on papers submitted after December 6th, 2016. Please ensure that you have an ORCID iD and that it is validated in Editorial Manager. To do this, go to ‘Update my Information’ (in the upper left-hand corner of the main menu), and click on the Fetch/Validate link next to the ORCID field. This will take you to the ORCID site and allow you to create a new iD or authenticate a pre-existing iD in Editorial Manager. Please see the following video for instructions on linking an ORCID iD to your Editorial Manager account: https://www.youtube.com/watch?v=_xcclfuvtxQ

Reviewers' comments:

Reviewer's Responses to Questions

**Comments to the Author**

1. Is the manuscript technically sound, and do the data support the conclusions?

Reviewer #1: Yes

Reviewer #2: Partly

2. Has the statistical analysis been performed appropriately and rigorously? 

Reviewer #1: Yes

Reviewer #2: Yes

3. Have the authors made all data underlying the findings in their manuscript fully available?

Reviewer #1: Yes

Reviewer #2: Yes

4. Is the manuscript presented in an intelligible fashion and written in standard English?

Reviewer #1: Yes

Reviewer #2: No

5. Review Comments to the Author

Reviewer #1: The paper by Zhang entitled A meta-analysis of circulating microRNAs in the diagnosis of papillary thyroid

carcinoma covers an interesting subject. I suggest only minor text editing. In particular (line 37 page 2) the Authors affirms that "Less than 20% of surgically removed nodules are malignant". However, this statement it is correct only if it is referred to the FN/SFN Bethesda category. Indeed, I suggest to discuss in brief both the limit of FNA in particular in indeterminate categories and the use of miRNA in thyroid cytopathology specimens.

Reviewer #2: The manuscript entitled "A meta-analysis of circulating microRNAs in the diagnosis of papillary thyroid carcinoma" highlighted that circulating microRNAs may be promising molecular markers for the diagnosis of 22 papillary thyroid carcinoma.

- The manuscript may benefit from a language revision by an English native speaker.

- The Authors should provide better quality figures.

- The Authors should address how selection bias have been solved.

- The Authors should better discuss if there are any difference in the role of serum and plasma extracted miRNAs in the diagnosis of PTC.

- The Authors should provide the extensive forms for all acronyms through the text when they first appear.

6. PLOS authors have the option to publish the peer review history of their article (what does this mean?). If published, this will include your full peer review and any attached files.

Reviewer #1: No

Reviewer #2: No

---

## [Author Response · Author response to Decision Letter 0]

22 Mar 2021

Point-to-point Reply to the comments

1. Please ensure that your manuscript meets PLOS ONE's style requirements, including those for file naming. The PLOS ONE style templates can be found athttps://journals.plos.org/plosone/s/file?id=wjVg/PLOSOne_formatting_sample_main_body.pdfhttps://journals.plos.org/plosone/s/file?id=ba62/PLOSOne_formatting_sample_title_authors_affiliations.pdf

Reply: Thank you for your suggestions. In the revised manuscript, we modified the paper style include title, reference and so on to meet the PLOS One’s requirements. 

2. At this time, we ask that you please provide the full search strategy and search terms for at least one database used as Supplementary Information.

Reply: Thank you for your careful comments. Full search strategy and search terms of this manuscript were sumarized in the supplement file 1. 

3. Thank you for including the statement that "The database search was conducted on XXXXXX". Please revise this statement to clarify whether all databases were searched from inception, or if there were any limits placed on the publication dates in your search.

Reply: Thank you for your professional suggestions. We revised the statement and added the limits placed on the publication dates in page 4 line 71 to line 73 of the paper. 

4. Please ensure you have thoroughly discussed any potential limitations of this study within the Discussion section, including the potential impact of confounding factors.

Reply: Thank you for your consideration suggestion. We have revised the discussion section according to your suggestion in page 11 line 228 to line 232 and page 12 line 241 to line 245 as well as page 13 line 268 to line 274 in the revised paper

Reply: Thank you for your suggestions. This manuscrpt’s database were acquired from the published literatures, and the database were all showed in the article. Search strategy has been provided in the supporting data. At the same time, we deleted the Data Availability Statement at the end of the paper.

6. We note that the grant information you provided in the ‘Funding Information’ and ‘Financial Disclosure’ sections do not match. When you resubmit, please ensure that you provide the correct grant numbers for the awards you received for your study in the ‘Funding Information’ section.

Reply: We are so sorry for our negligence. In the revised version of the manuscript, we carefully re-check each Funding Information and revised them. But we can't change the information of fundings in the financial disclosure section. We have 5 fundings, which have been listed in detail at the end of the manuscript. However, only 4 fundings are displayed in the financial disclosure section, and we can't find a dialog box that can be modified. We would be greatly appreciated you for your consideration to help me update the information in financial disclosure section. We have request the updating of fundings in cover letter.

7. PLOS requires an ORCID iD for the corresponding author in Editorial Manager on papers submitted after December 6th, 2016. Please ensure that you have an ORCID iD and that it is validated in Editorial Manager. To do this, go to ‘Update my Information’ (in the upper left-hand corner of the main menu), and click on the Fetch/Validate link next to the ORCID field. This will take you to the ORCID site and allow you to create a new iD or authenticate a pre-existing iD in Editorial Manager. Please see the following video for instructions on linking an ORCID iD to your Editorial Manager account: https://www.youtube.com/watch?v=_xcclfuvtxQ

Reply: Thank you for your suggestions. I have registered ORCID ID number in your magazine, it is 0000-0002-9729-8995. I have updated my Information and authenticated the pre-existing iD in Editorial Manager, now, it has been validated in my Editorial Manager account. 

8. Reviewer #1: The paper by Zhang entitled A meta-analysis of circulating microRNAs in the diagnosis of papillary thyroidcarcinoma covers an interesting subject. I suggest only minor text editing. In particular (line 37 page 2) the Authors affirms that "Less than 20% of surgically removed nodules are malignant". However, this statement it is correct only if it is referred to the FN/SFN Bethesda category. Indeed, I suggest to discuss in brief both the limit of FNA in particular in indeterminate categories and the use of miRNA in thyroid cytopathology specimens.

Reply: Thank you for your insightful comments. According to your opinion, we discussed the the limit of FNA in indetermination particular categories and the use of miRNA in thyroid cytopathology specimens in page 11-12 from line 241 to line 245.

9. Reviewer #2: The manuscript entitled "A meta-analysis of circulating microRNAs in the diagnosis of papillary thyroid carcinoma" highlighted that circulating microRNAs may be promising molecular markers for the diagnosis of papillary thyroid carcinoma.

9.1 The manuscript may benefit from a language revision by an English native speaker.

Reply: Thank you for your suggestions. The language of this manuscript was revised by an English native speaker. We hope that the revisions in the current manuscript will be sufficient for consideration of publication in ‘‘PLOS ONE

9.2 The Authors should provide better quality figures.

Reply: Thank you for your suggestions. In the revised version, we resubmit the new figures with higher resolution, we hope that it can meet the requirements of “PloS one”

9.3 The Authors should address how selection bias have been solved.

Reply: Thanks for your professional suggestion.9 literatures including 29 studies were analyzed in this study. The literatures were from Asian and European, and the heterogeneity test shows the I2 is very high (84%~90%). Higher heterogeneity which may be a selection bias, and may can influence the results, we used subgroup analysis to reduce the heterogeneity. The details are in the section of “subgroup analysis”. 

9.4 The Authors should better discuss if there are any differences in the role of serum and plasma extracted miRNAs in the diagnosis of PTC.

Reply: Thanks for your thoughtful suggestion. We have revised the discussion section according to your suggestion in page 13 line 268 to line 274 in the revised paper

9.5 The Authors should provide the extensive forms for all acronyms through the text when they first appear. 

Reply: Thank you for your careful comments. We have provided the extensive forms for all acronyms through the text when they first appear through the whole revised paper according to your suggestion.

---

## [Decision Letter · Decision Letter 1]

30 Apr 2021

A meta-analysis of circulating microRNAs in the diagnosis of papillary thyroid carcinoma

PONE-D-20-40536R1

Dear Dr. Huan Xin Zhang

We’re pleased to inform you that your manuscript has been judged scientifically suitable for publication and will be formally accepted for publication once it meets all outstanding technical requirements.

Kind regards,

Giancarlo Troncone

Academic Editor

PLOS ONE

Additional Editor Comments (optional):

Reviewers' comments:

Reviewer's Responses to Questions

**Comments to the Author**

1. If the authors have adequately addressed your comments raised in a previous round of review and you feel that this manuscript is now acceptable for publication, you may indicate that here to bypass the “Comments to the Author” section, enter your conflict of interest statement in the “Confidential to Editor” section, and submit your "Accept" recommendation.

Reviewer #1: All comments have been addressed

2. Is the manuscript technically sound, and do the data support the conclusions?

Reviewer #1: Yes

3. Has the statistical analysis been performed appropriately and rigorously? 

Reviewer #1: I Don't Know

4. Have the authors made all data underlying the findings in their manuscript fully available?

Reviewer #1: Yes

5. Is the manuscript presented in an intelligible fashion and written in standard English?

Reviewer #1: Yes

6. Review Comments to the Author

Reviewer #1: I have no further comments to make. The Authors have addressed all the points raised by the reviewers.

7. PLOS authors have the option to publish the peer review history of their article (what does this mean?). If published, this will include your full peer review and any attached files.

Reviewer #1: No

---

## [Editor Report · Acceptance letter]

12 May 2021

PONE-D-20-40536R1 

A meta-analysis of circulating microRNAs in the diagnosis of papillary thyroid carcinoma 

Dear Dr. Zhang:

I'm pleased to inform you that your manuscript has been deemed suitable for publication in PLOS ONE. Congratulations! Your manuscript is now with our production department. 

Kind regards, 

on behalf of

Professor Giancarlo Troncone 

Academic Editor

PLOS ONE